# Complications of Port-a-Cath Systems: An Institutional Study on Romanian Oncological Patients

**DOI:** 10.3390/cancers18010174

**Published:** 2026-01-05

**Authors:** Adina Nemeș, Sebastian-Alexandru Pocol, Tunde Banciu, Diana Voskuil-Galoș

**Affiliations:** 1Medical Oncology Department, Faculty of Medicine, Iuliu Hațieganu University of Medicine and Pharmacy, 40012 Cluj-Napoca, Romania; adina.nemes@umfcluj.ro; 2Medical Oncology Department, The Oncology Institute Prof. Dr. Ion Chiricuță Cluj-Napoca, 40015 Cluj-Napoca, Romania; 3Faculty of Pharmacy, Iuliu Hațieganu University of Medicine and Pharmacy, 40012 Cluj-Napoca, Romania

**Keywords:** central venous catheters, port-a-cath systems, catheter-related infection, catheter thrombosis, oncological patient care

## Abstract

The present study assesses complications associated with port-a-cath systems (PCSs) in cancer patients. These devices are essential for drug administration in oncological care settings; however, they pose certain risks, such as infection, thrombosis, and other mechanical issues. By analyzing 124 patients at The Oncology Institute “Prof. Dr. Ion Chiricuță,” Cluj-Napoca, Romania, the authors aimed to determine the incidence of PCS-related complications, their relationship with known prognostic factors, and strategies for management. The findings show that PCSs are largely safe, with infections and thrombosis being the most common complications, most of which can be managed without device removal. Standardized care and early complication management facilitate long-term PCS use, potentially improving patient quality of life. These results provide real-world evidence for healthcare practitioners, encouraging safer PCS use and calling for future studies on complication prevention and management.

## 1. Introduction

Cancer represents a major public health problem, with millions of new cases being diagnosed each year, representing one of the leading causes of death worldwide according to the World Health Organization [1]. Venous access plays a pivotal role in oncological care, as specific oncological treatments, such as chemotherapy, immunotherapy, and targeted therapy, require intravenous (IV) administration. Moreover, best supportive care protocols may additionally call for drug delivery via a parenteral route [2]. Central venous catheters (CVCs), such as peripherally inserted central catheters (PICCs) or port-a-cath systems (PCSs), can be used in oncological patients who have to undergo IV treatments for a long period of time, but these types of catheters can be subjected to complications, some even resulting in high morbidity and mortality [3,4]. CVC complications can be categorized as mechanical, infectious, or thrombotic, each with specific prognostic and therapeutic implications. Such adverse outcomes may determine repeated hospitalizations, treatment interruption, and warrant device replacement [5,6,7]. Furthermore, they can represent major sources of anxiety, insecurity, and distrust in treatment for the patient. Care monitoring, patient education, and adherence to care protocols are essential in preventing numerous complications associated with CVCs [7,8,9]. The aim of this study was to evaluate PCS-associated complications and their management in oncological patients treated at The Oncology Institute “Prof. Dr. Ion Chiricuță,” Cluj-Napoca, Romania.

## 2. Materials and Methods

This study was a non-randomized, observational, retrospective study that included patients with histologically confirmed cancer who had a PCS implanted at The Oncology Institute “Prof. Dr. Ion Chiricuță,” Cluj-Napoca, between 1 January 2024 and 31 December 2024. The study design allowed us to thoroughly evaluate and record complications related to the PCS. All patients included in this study signed a written informed consent. The Ethics Committee of The Oncology Institute “Prof. Dr. Ion Chiricuță,” Cluj-Napoca, evaluated and approved the study design.

The main objectives of this study were the evaluation of the incidence of PCS-related complications, the analysis of complication occurrence in relation to a series of known prognostic factors, and the identification of those complications that were significantly associated with PCS removal.

This study included adult patients (age > 18 years) with histologically confirmed cancer who had a PCS implanted. The patients’ medical records were retrospectively evaluated in order to confirm the availability of baseline information: the surgical protocol during the implantation of the central system, in addition to standard discharge documentation. All patients were required to adhere to a personalized follow-up protocol. The personnel responsible for the follow-up protocol were represented by physicians and nurses. The nurses were instructed to report any clinical signs related to the PCS or any port malfunction to the physician. For the patients who underwent intravenous systemic treatment, the functionality and possible complications of the PCS were evaluated at every treatment administration (usually 14 or 21 days). Every time the PCS was accessed and the needle was inserted, the functionality of the system was checked, and a saline solution was flushed. During treatment, a saline solution was flushed between two drugs that were administered. If the PCS was not accessed for a minimum of two weeks, lock therapy with 5 mL of heparinized serum 100 IU/mL was administered. In the patients who no longer underwent intravenous systemic treatment, follow-up was performed every 6 weeks, and the PCS was heparinized with 5 mL of heparinized serum 100 IU/mL every 6 weeks. When accessing the PCS, aseptic techniques were followed, including local asepsis with betadine or chlorhexidine, sterile gloves, disinfection of connections with chlorhexidine, and transparent sterile dressings. The PCS dressing was inspected every day, and when any changes were observed, the dressing was replaced. If there were no changes in the dressing, it was changed every 7 days. When the needle was removed, a sterile adhesive dressing was applied. The patients were shown how to visually inspect the port every day and asked to contact the physician if there were any changes, pain, or erythema.

Regarding the time of occurrence of PCS complications, these were classified as early-onset complications (occurring within ≤30 days of port-a-cath system implantation) and late-onset complications (occurring within >30 days of port-a-cath system implantation). The patients included must have had available medical documents that allowed the evaluation of the data necessary for the completion of this research. Patients with incomplete or missing data from the medical records or patients who did not adhere to the follow-up protocol were excluded from the study database and further statistical case analysis. An additional basis for exclusion was represented by patients with contraindications for the implantation of a PCS, particularly trauma to the cervical region, history of radiotherapy to the cervical or upper thoracic region, known venous path abnormalities, severe coagulopathies, severe thrombocytopenia, bulky mediastinal tumors, or superior vena cava compression syndrome.

The following information was collected for each patient: demographic data (age, sex), clinical information (cancer type and stage), technical details (vein used for implantation—right/left subclavian vein), associated complications, both local and systemic (postprocedural bleeding/hematoma, surgical wound dehiscence, catheter-related venous thrombosis, catheter infections—local or systemic, chemotherapy extravasation, catheter migration/dislocation, inability to use the port immediately after implantation, rotation/torsion of the port in the subcutaneous plane, sepsis), treatment, and management of the complications. PCS-related infection was diagnosed by collecting samples from purulent secretions when present and blood cultures for both aerobes and anaerobes, simultaneously from the PCS and from peripheral veins. PCS-related thrombosis was diagnosed by CT angiography. Additionally, the requirement for permanent removal of the PCS was documented, whether or not it was subsequently followed by implantation of a new system at the same or a different anatomical site.

The data collected from the patients’ medical records were used to create a Microsoft Excel database. Quantitative variables were expressed as mean ± standard deviation (SD). Their comparison was performed using an independent samples *t*-test for normally distributed data and a Mann–Whitney U-test for data that did not meet the normality assumption (verified by the Shapiro–Wilk test). Qualitative variables were reported as absolute values and percentages. The following were used for their comparison: the Chi-square (Chi^2^) test and the Fisher-exact test, in situations where more than 20% of the theoretical frequencies were <5. For all tests, statistical significance was considered at a *p*-value less than 0.05.

## 3. Results

A total of 142 oncological patients with a PCS implanted at The Oncology Institute “Prof. Dr. Ion Chiricuță,” Cluj-Napoca, between 1 January 2024 and 31 December 2024 were initially included in this study. Out of the 142 patients, 18 patients were excluded due to a lack of adequate postoperative follow-up. After the exclusion of ineligible patients, 124 patients who met the inclusion criteria and for whom complete postoperative follow-up data were available were included in the final analysis.

The median follow-up for the patients included in this study was 12 months (1–19 months). The mean age of the patients was 58 years (range 19–81 years), and most patients were female, 64% (79 patients).

The analysis of the cancer type and stage showed that 44% (54 patients) of the patients had colorectal cancer, followed by gastric cancer (12%), breast cancer (11%), ovarian cancer (10%), pancreatic cancer (7%), endometrial cancer (4%), lung cancer (4%), and testicular cancer (3%) (Table 1). Regarding cancer stage, 61% of the patients (75 patients) were diagnosed with stage IV disease, while 2% (3 patients) had stage I, 8% (10 patients) had stage II, and 29% (36 patients) had stage III (Figure 1).

In 89% of the patients, the PCS was implanted in the right subclavian vein, while in 11%, it was placed in the left subclavian vein.

Complications related to the PCS were observed in 20% of the patients included in this study (25 patients). Most patients presented only one complication, while two patients experienced two simultaneous complications. The mean interval in which the complications occurred was 47 days (0–487 days). Early-onset complications were observed in 40% of the patients, with a mean interval of occurrence of 15 days. Sixty percent of the patients presented late-onset complications, with a mean interval of occurrence of 115 days.

When analyzing the incidence of complications in relation to the side where the PCS was implanted (right side vs. left side), 36% of the patients with a PCS implanted in the left subclavian vein presented complications vs. 18% of the patients with a PCS implanted in the right subclavian vein, but this association was not statistically significant (*p* = 0.155) (Figure 2).

The most frequent complication observed was PCS infection, occurring in 10 patients, followed by surgical wound dehiscence and thrombosis, each reported in 6 patients. Bleeding, extravasation, catheter migration, inability to use the port, and torsion were each observed in one patient (Table 2). For infection, the incidence rate per 1000 catheter-days was 5.93.

Overall, 90% of the patients (nine patients) who presented with an infection as a PCS-related complication were diagnosed with colorectal cancer, and one patient was diagnosed with breast cancer. Among the patients who presented thrombosis as a complication, 50% (three patients) were diagnosed with colorectal cancer and 50% (three patients) with breast cancer.

Out of the 10 patients who presented with a PCS infection, in 8 patients, only one bacterial species was isolated. Two patients had polymicrobial infections, and one patient additionally had *Candida crusei*. The most frequently isolated germ was *Staphylococcus aureus* (found in five patients), while other germs (*Escherichia coli*, *Klebsiella aerogenes*, *Stenotrophomonas maltophilia*) were each isolated in one patient. Treatment with antibiotics was administered to all patients with a PCS infection: five patients received monotherapy, and five patients received an association of antibiotics. Four percent of the patients with an infectious complication of the PCS developed sepsis; however, no deaths due to sepsis were reported.

All six patients with thrombosis of the PCS received fibrinolytic treatment with IV Taurolock urokinase 25,000 IU. In five patients, the catheter was re-permeabilized, and in one patient, the catheter remained nonfunctional and therefore had to be removed.

In 12% of the cases (three patients), the PCS had to be removed due to the complications that occurred, and in 10% of the patients, the catheter was re-implanted.

When analyzing the occurrence of PCS complications, i.e., early and late complications, in relation to a series of prognostic factors, such as sex, age, cancer type, stage of the disease, implantation site, and type of complication, there was no statistical significance for any of the prognostic factors analyzed. There was a tendency for statistical significance for thrombosis (*p* = 0.051) (Table 3).

## 4. Discussion

The analysis of patients who had a PCS implanted at The Oncology Institute “Prof. Dr. Ion Chiricuță,” Cluj-Napoca, between 1 January 2024 and 31 December 2024 revealed a 20% incidence of complications, aligning with previously published data. The incidence of PCS-related complications reported in the literature varies between 10% and 30%, depending on the study population, implantation technique, and care protocols [7,10,11]. In a study that included 1716 patients, Li et al. showed a PCS-related complication incidence of 18.5%, which is similar to the incidence reported in our study [7].

The majority of the patients included in our study were female patients (64%) diagnosed with digestive tract tumors and breast cancer, who had a mean age of 58 years.

Only 11% of the patients had the PCS implanted in the left subclavian vein, but the percentage of complications was double in these patients compared to the patients who had the PCS implanted in the right subclavian vein (36% vs. 18%), although the difference remained without statistical significance. Some studies suggest that left subclavian vein PCS implantation is associated with a higher rate of complications and prolonged procedure time, mainly due to the anatomy and distance to the superior vena cava [12,13,14]. In a study of 550 patients, Ignatov et al. showed that the highest incidence of complications was observed in patients with a totally implanted central venous access port placed in the left subclavian vein or with the catheter tip localized in the peripheral part of the superior vena cava [13]. In a study of 1848 patients, catheter occlusion was statistically more frequent when the PCS was implanted in the left subclavian vein versus the right subclavian vein (3.5% vs. 1%, *p* = 0.001) [14].

PCS complications occurred at a mean interval of 47 days after implantation. Most patients in our study (60%) presented late-onset complications, with a mean interval of 115 days. The large interval for complication occurrence observed in our study (0–487 days) and in other studies (3–1872 days) suggests the need for continuous follow-up, extending beyond the initial few months when we would expect a higher risk of complications [15].

PCS infection was the most frequent complication observed, affecting 8% of the patients, and it is the most frequent complication reported in studies that evaluated PCS complications, which is consistent with the data published in the literature (3.2–20%) [4,7,16,17,18]. Gram-positive bacteria were the most frequently isolated germs, notably *Staphylococcus aureus*, but there were also polymicrobial infections identified in addition to fungi (*Candida crusei*). A systematic review published by Raad et al. confirms that *Staphylococcus aureus* is the most frequent pathogen isolated, followed by coagulase-negative staphylococci and *Candida saprophyticus* [17]. Other studies also report that the most frequent pathogen isolated in patients with PCS-related infection is *Staphylococcus aureus* [19,20,21]. All these data emphasize the importance of hygiene and antisepsis measures when manipulating PCSs. Fortunately, only 4% of the patients with an infectious complication developed sepsis, and none of these cases resulted in death.

Thrombosis of the PCS occurred exclusively as a late-onset complication, with a tendency towards statistical significance, which may reflect a gradual accumulation of prothrombotic factors in oncological patients in addition to the limitations of antithrombotic prophylaxis. Oncological patients are at increased risk of thrombus formation due to the prothrombotic state associated with malignancy [22]. In addition to cancer-associated thrombosis susceptibility, during the insertion of the catheter, the endothelium of the vein may suffer mechanical injury [23]. Furthermore, the presence of the catheter in the vein and its use for drug administration causes stasis [24]. The association between these three factors, i.e., hypercoagulability, endothelial injury, and stasis, also known as Virchow’s triad, explains the physiopathology of thrombus formation in the context of PCSs [24]. However, there are certain factors related to catheter insertion that could also increase the risk of thrombosis, notably, left-sided insertion, cases of multiple insertion attempts, or cases where the tip of the catheter remains located above the cavo-atrial junction [24]. In oncological patients, the risk of catheter-associated thrombosis is additionally increased in the setting of sepsis and exposure to specific chemotherapeutic drugs [25]. In addition to the catheter-related factors and the thrombogenic predisposition associated with malignancy, patients with certain cancer types may have an increased risk of PCS-related thrombosis. One small study identified a correlation between catheter thrombosis and specific cancer locations, particularly colon, breast, and kidney [26]. Another study identified breast cancer as a tumor site associated with an increased thrombotic risk and reported bladder cancer as having the strongest association with catheter-related thrombosis [27]. A study conducted on 600 patients suggested that gastric and pancreatic cancers were significantly associated with thrombosis occurrence [28]. Moreover, given the well-established hypercoagulable state associated with pancreatic cancer, one study reported an incidence of catheter-related thrombosis of nearly 15% in patients with pancreatic cancer [29]. In our study, half of the patients who presented with catheter thrombosis as a PCS-related complication were diagnosed with colorectal cancer, and half with breast cancer. However, despite these well-supported correlations between cancer type and catheter thrombosis, the small number of patients with thrombosis in the current study does not allow us to suggest a direct link between tumor location and subsequent complications. Nevertheless, large-scale studies are required in order to further evaluate such clinical associations.

Literature data also indicate that PCS-related thrombosis is a common late complication, typically occurring after 30 days and often presenting with mild clinical signs, such as ipsilateral limb or neck pain and swelling, while a proportion of cases remain asymptomatic [8,9]. The pathophysiological mechanisms of catheter-related thrombosis may explain its delayed occurrence, as prolonged catheter presence promotes vessel hyperplasia and thrombus stabilization. In addition, catheter colonization and infection could further potentiate thrombosis through fibrin sheath formation and persistent thrombin activity [9]. The incidence of PCS-related thrombosis varies between 2.8% and 12.8%, and it is lower than the incidence presented by PICC [30,31,32]. Our study noted a 24% incidence of catheter thrombosis. The variability in reported incidence rates may be attributed to differences in study design and endpoints, as well as heterogeneity in patient populations, cancer types, and additional risk factors across studies.

Even though PCS thrombosis can be a life-threatening complication, it is usually resolved with urokinase and rarely requires catheter removal. In our study, treatment with IV Taurolock urokinase 25,000 IU led to the re-permeabilization of the catheter in 83% of the patients. Removal of the catheter due to complications was necessary in three patients, and in 10% of the patients, the PCS was re-implanted, results comparable to those found in the existing literature [7,15,33].

Even though this research provides a real-world picture of PCS-related complications in oncological patients, it is important to acknowledge the limitations of this study. The retrospective nature of this study implies an increased risk of selection bias and a lack of control over confounding factors, such as insertion technique, level of experience of the personnel, or adherence to PCS care protocols. In addition, this was a non-randomized study conducted in a single institution. The limited number of patients included in this study, especially patients with PCS-related complications, weakens the statistical power of the analysis and may not present a complete characterization of the general population evaluated in this study [34,35,36].

## 5. Conclusions

This study reflects our single-center experience with PCSs in oncological patients and provides real-world evidence regarding the relative safety of PCSs. Although the small sample size represents a significant limitation and may restrict the broader applicability of the results in other clinical settings, our findings suggest that the complication profile of PCSs is mainly dominated by infections and thrombotic events. However, despite a non-negligible incidence of such adverse events, the majority of complications were successfully managed conservatively with antibiotic and fibrinolytic therapy, without requiring catheter removal. However, in 12% of the cases, catheter removal was required. For clinicians, these results highlight the importance of careful surveillance, early recognition, and early management of PCS-related complications. Moreover, adherence to standardized implantation techniques and standardized catheter care protocols remains crucial in minimizing complications and allowing prolonged PCS use, thus improving the quality of life of cancer patients. Dedicated personnel must be trained periodically about the flushing and locking protocols and about the correct aseptic techniques and dressing-change procedures required for accessing the PCS. Patients should also be educated about the clinical signs that can occur and instructed to immediately contact their physician.

## Figures and Tables

**Figure 1 cancers-18-00174-f001:**
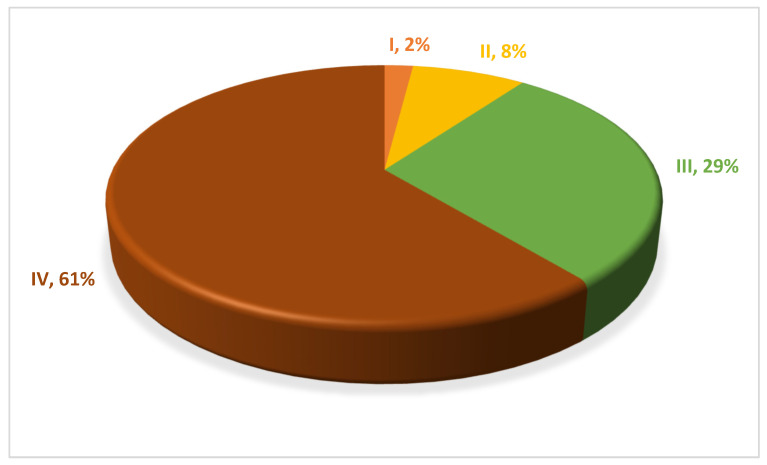
Cancer stage among patients with PCS. Cancer stage (I—stage I, II—stage II, III—stage III, IV—stage IV) and percentage of patients with each cancer stage among our study population.

**Figure 2 cancers-18-00174-f002:**
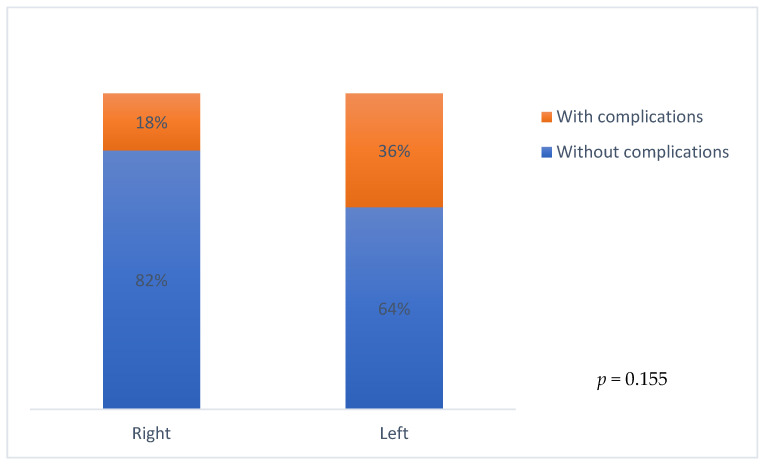
Distribution of patients who developed complications according to PCS location. This figure shows the incidence of complications (percentage) in relation to where the PCS was implanted (right subclavian vein vs. right subclavian vein). Analysis between implantation in the right vs. left subclavian vein showed no statistical significance (*p* = 0.155).

**Table 1 cancers-18-00174-t001:** Cancer type in the studied patient population.

Cancer Type	Number of Patients	%
Colorectal cancer	54	44
Gastric cancer	15	12
Breast cancer	13	10
Ovarian cancer	12	10
Pancreatic cancer	9	7
Endometrial cancer	5	4
Lung cancer	5	4
Testicular cancer	4	3
Bone cancer	2	2
Small intestine cancer	1	0.81
Head and neck cancer	1	0.81
Cervical cancer	1	0.81
Central nervous system cancer	1	0.81
Malignant melanoma	1	0.81
Total = 124 patients		

**Table 2 cancers-18-00174-t002:** PCS complications in our study.

PCS Complication	Number of Patients
Infection	10
Surgical wound dehiscence	6
Catheter thrombosis	6
Bleeding	1
Extravasation	1
Catheter migration	1
Inability to use the port	1
Catheter torsion	1

**Table 3 cancers-18-00174-t003:** PCS complication occurrence in relation to sex, age, cancer type, stage of the disease, implantation site, and type of complication.

	All Patients (*n* = 25)	Early Complications (*n* = 10)	Late Complications (*n* = 15)	*p*
Sex				0.075
M	7 (28%)	5 (50%)	2 (13%)	
F	18 (72%)	5 (50%)	13 (87%)	
Age	56.56 ± 15.92	57 ± 17.23	56.27 ± 15.6	0.913
Cancer type				0.694
Digestive tract	16 (64%)	7 (70%)	9 (60%)	
Gynecological	7 (28%)	3 (30%)	4 (27%)	
Testicular	1 (4%)	0 (0%)	1 (6.67%)	
Lung	0 (0%)	0 (0%)	1 (6.67%)	
Other	1 (4%)	0 (0%)		
Cancer stage				0.504
I	1 (4%)	1 (10%)	0 (0%)	
II	2 (8%)	1 (10%)	1 (7%)	
III	6 (24%)	3 (30%)	3 (20%)	
IV	16 (64%)	5 (50%)	11 (73%)	
Catheter location				0.358
Right	20 (80%)	7 (70%)	13 (87%)	
Left	5 (20%)	3 (30%)	2 (13%)	
Bleeding/hematoma	1 (4%)	1 (10%)	0 (0%)	0.399
Surgical wound dehiscence	6 (24%)	3 (30%)	3 (20%)	0.653
Catheter thrombosis	6 (24%)	0 (0%)	6 (40%)	0.051
Infection	10 (40%)	4 (40%)	6 (40%)	0.999
Extravasation	1 (4%)	0 (0%)	1 (6.67%)	0.999
Catheter migration	1 (4%)	1 (10%)	0 (0%)	0.399
Inability to use the port	1 (4%)	1 (10%)	0 (0%)	0.399
Catheter torsion	1 (4%)	0 (0%)	1 (6.67%)	0.999

## Data Availability

Data are contained within this article.

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
