# Peer review of "Complications of Port-a-Cath Systems: An Institutional Study on Romanian Oncological Patients"

_cancers, 2026, doi:10.3390/cancers18010174_

Round 1
Reviewer 1 Report
Comments and Suggestions for Authors
Authors evaluated PCS associated complications and their management in oncological patients treated in The Oncology Institute.
Although this manuscript is well written, several issues arise.
- This study size is small. If possible, If possible, the author should increase the number of cases and resubmit.
- Authors should explain results of figure in Figure legend.
- Conclusion did not show useful information in clinician.
- The complication rate related to the PCS was 20%. I think that 2o% is too high.
- In the conclusion, it is mentioned that removal is rare, but what percentage was actually removed?
- Which complications are related to the parts of PCS? Figure 2 needs further information.
- Proposal for reduced PCS complication would be required.
- What causes catheter thrombosis?
Author Response
Letter to reviewer 1
Manuscript ID: cancers-4049073
Complications of port-a-cath systems: an institutional study on Romanian oncological patients
Adina NemeÈ™ 1,2, Sebastian-Alexandru Pocol 3, Tunde Banciu 2 and Diana Voskuil-GaloÈ™ 1,2,*
Reviewer 1 Comments and Suggestions for Authors – round 1
- This study size is small. If possible, If possible, the author should increase the number of cases and resubmit.
- Authors should explain results of figure in Figure legend.
- Conclusion did not show useful information in clinician.
- The complication rate related to the PCS was 20%. I think that 2o% is too high.
- In the conclusion, it is mentioned that removal is rare, but what percentage was actually removed?
- Which complications are related to the parts of PCS? Figure 2 needs further information.
- Proposal for reduced PCS complication would be required.
- What causes catheter thrombosis?
1.This study size is small. If possible, If possible, the author should increase the number of cases and resubmit.
We thank the reviewer for the time the time assigned to analyse our manuscript. We agree that a larger sample size would strengthen the results of the study. However, the number of cases cannot be increased in a timely manner, as it would require significant resources to evaluate the data obtained from the additional patients and conclude statistically relevant results.
To address this limitation, we have now explicitly acknowledged the small sample size in the Limitations and Conclusion sections of the article and discussed the potential impact on the findings that we, the authors, obtained. Despite the limited number of cases, the study provides valuable preliminary evidence and detailed analysis of the complications related to port-a-cath systems, encouraging future larger studies.
- Authors should explain results of figure in Figure legend.
We thank the reviewer for the suggestion. Figure one legend has been updated and includes the explanation of the results
- Conclusion did not show useful information in clinician.
We appreciate the reviewer’s comment. Please see below the changes we have made in order to increase the impact of the Conclusion section:
This study reflects our single-center experience with PCS in oncological patients and provides real-world evidence regarding the relative safety of PCS. Although the small sample size represents a significant limitation and may restrict the broader applicability of the results in other clinical settings, our findings suggest that the complication profile of PCS is mainly dominated by infections and thrombotic events. However, despite a non-negligible incidence of such adverse events, the majority of complications were successfully managed conservatively with antibiotic and fibrinolytic therapy, without requiring catheter removal. However, in 12% of the cases, catheter removal was required. For clinicians, these results highlight the importance of careful surveillance, early recognition, and early management of PCS-related complications. Moreover, adherence to standardized implantation techniques and standardized catheter care protocols remain crucial in minimizing complications and allowing prolonged PCS use, thus improving the quality of life of cancer patients.
4.The complication rate related to the PCS was 20%. I think that 2o% is too high.
We appreciate the comment of the reviewer. The complication rate obtained as a result of our data analysis aligns with previously reported complication rates that vary between 10% and 30%. However, we acknowledge that the observed complication rate of 20% may also be influenced by the relatively small sample size, which could lead to an increased impact of individual events when taking into consideration estimations expressed in percentage.
5.In the conclusion, it is mentioned that removal is rare, but what percentage was actually removed?
We thank the reviewer for the suggestion to include the percentage of patients requiring PCS removal. We have amended the conclusion to include this important result.
- Which complications are related to the parts of PCS? Figure 2 needs further information.
We appreciate the suggestion of the reviewer. Unfortunately we did not analyse the occurrence of complications in relation to the parts of the PCS (port or catheter). Further information was added to figure 2 legend.
- Proposal for reduced PCS complication would be required.
We appreciate the reviewer’s comment. The main aim of this article was to evaluate the incidence of PCS-related complications and to analyse the most common complications. As per your comment please find bellow a paragraph including some recommendations for reducing PCS complications that was included in the Conclusion section.
Dedicated personnel must be trained periodically about the flushing and locking protocols and about the correct aseptic techniques and dressing-change procedures required for accessing the PCS. Patients should also be educated about the clinical sings that can occur and instructed to immediately announce their physician.
8.What causes catheter thrombosis?
We value the suggestion to include information about the causes of catheter thrombosis. Please find below the paragraph explaining the physiopathology of thrombus formation in the context of port-a-cath systems.
Oncological patients are at increased risk of thrombus formation, due to the prothrombotic state associated with malignancy. In addition to the cancer-associated thrombosis susceptibility, during the insertion of the catheter, the endothelium of the vein may suffer mechanical injury . Furthermore, the presence of the catheter in the vein and its use for drug administration causes stasis. The association between these three factors, hypercoagulability, endothelial injury and stasis, also known as the Virchow’s triad, explains the physiopathology of thrombus formation in the context of PCS. However, there are certain factors related to the catheter insertion that could also increase the risk of thrombosis, notably the left sided insertion, cases of multiple insertion attempts or cases where the tip of the catheter remains located above the cavo-atrial junction (3). In oncological patients, the risk of catheter-associated thrombosis is additionally increased in the setting of sepsis and exposure to specific chemotherapeutic drugs.
Reviewer 2 Report
Comments and Suggestions for Authors
The topic is clinically relevant, and the reported findings are broadly consistent with the 10–30% complication rate of PCS described in the literature. I recommend that the authors complete and report key methodological and results items in accordance with the STROBE checklist before the manuscript can be considered for acceptance.
- The manuscript mentions excluding cases due to “incomplete medical records” and/or “failure to adhere to follow-up,” but it does not report the number of excluded patients or the distribution of exclusion reasons. This omission may introduce selection bias and should be addressed.
- The authors report isolated pathogens and antimicrobial therapy for infections; however, essential operational details are missing, including how PCS-related infection was diagnosed/defined, sampling procedures, and microbiological culture criteria. Similarly, “thrombosis” should be clearly defined and the imaging modality or diagnostic criteria used for confirmation should be specified. Without standardized definitions, inter-center comparisons are difficult.
- The results are currently reported mainly as patient-level proportions and an “average time to complication of 47 days (0–487 days).” PCS-related complications would be more appropriately presented as incidence rates per 1,000 catheter-days, and the manuscript should also provide the median catheter dwell time and/or follow-up duration to facilitate comparison with prior studies.
- The Methods state that all patients “must follow an individualized follow-up schedule,” but key details are not provided, including follow-up frequency, responsible personnel, flushing/locking protocols, aseptic technique, dressing-change procedures, and patient education. These factors are critical determinants of infection and occlusion risk and should be explicitly described.
- The current analysis primarily compares “early” versus “late” complications, and Table 3 performs subgroup comparisons only within the complication cases (N=25), which substantially limits statistical power. The borderline result for thrombosis (p = 0.051) should be interpreted cautiously and should not be overstated.
- The study period is 2024, whereas the ethics approval date is 2025, and the manuscript states that all patients provided informed consent. The authors should clarify whether this is a prospective follow-up cohort or a retrospective chart review. If retrospective, they should state whether consent was waived, obtained retrospectively, or otherwise addressed according to institutional requirements.
- The manuscript contains multiple spelling and grammatical errors; comprehensive language editing is recommended.
Author Response
Letter to reviewer 2
Manuscript ID: cancers-4049073
Complications of port-a-cath systems: an institutional study on Romanian oncological patients
Adina NemeÈ™ 1,2, Sebastian-Alexandru Pocol 3, Tunde Banciu 2 and Diana Voskuil-GaloÈ™ 1,2,*
Reviewer 2 Comments and Suggestions for Authors – round 1
- The manuscript mentions excluding cases due to “incomplete medical records” and/or “failure to adhere to follow-up,” but it does not report the number of excluded patients or the distribution of exclusion reasons. This omission may introduce selection bias and should be addressed.
We thank the reviewer for the time the time assigned to analyze our manuscript and for the suggestion. Information about the excluded cases was introduced in the Results paragraph.
There were 142 oncological patients with a PCS implanted in The Oncology Institute „Prof. Dr. Ion Chiricuță” Cluj-Napoca between 01.01.2024-31.12.2024 initially included in this study. Out of the 142 patients, 18 patients were excluded due to lack of adequate postoperative follow-up. After the exclusion of ineligible patients, 124 patients who met the inclusion criteria and for whom complete postoperative follow-up data were available were included in the final analysis.
- The authors report isolated pathogens and antimicrobial therapy for infections; however, essential operational details are missing, including how PCS-related infection was diagnosed/defined, sampling procedures, and microbiological culture criteria. Similarly, “thrombosis” should be clearly defined and the imaging modality or diagnostic criteria used for confirmation should be specified. Without standardized definitions, inter-center comparisons are difficult.
We value the suggestion to include information about the diagnosis of PCS-related infection and thrombosis. Please find below the paragraph with this information that was included in the Materials and methods section.
PCS-related infection was diagnosed by collecting samples from purulent secretions if they were present and blood cultures for both aerobes and anaerobes simultaneously from the PCS and from peripheral veins. PCS-related thrombosis was diagnosed by CT angiography.
- The results are currently reported mainly as patient-level proportions and an “average time to complication of 47 days (0–487 days).” PCS-related complications would be more appropriately presented as incidence rates per 1,000 catheter-days, and the manuscript should also provide the median catheter dwell time and/or follow-up duration to facilitate comparison with prior studies.
We thank the reviewer for the suggestion. The suggested information was included in the Results section.
The median follow-up for the patients included in this study was 12 months (1-19 months).
For infection the incidence rate per 1000 catheter-days was 5.93.
- The Methods state that all patients “must follow an individualized follow-up schedule,” but key details are not provided, including follow-up frequency, responsible personnel, flushing/locking protocols, aseptic technique, dressing-change procedures, and patient education. These factors are critical determinants of infection and occlusion risk and should be explicitly described.
We appreciate the suggestion of the reviewer. Please find bellow the follow-up protocol that was used in our study and included in the Materials and methods section.
The personnel responsible for the follow-up protocol was represented by physicians and nurses. Nurses were instructed to report any clinical signs related to the PCS or any port malfunction to the physician. For patients who underwent intravenous systemic treatment the functionality and possible complications of the PCS were evaluated at every treatment administration (usually 14 or 21 days). Every time the PCS was accessed and the needle was inserted the functionality of the system was checked and saline solution was flushed. During treatment saline solution was flushed between two drugs that were administered. If the PCS was not accessed for minimum two weeks lock therapy with 5ml of heparinized serum 100U/ml was administered. In patients who no longer underwent intravenous systemic treatment follow-up was performed every 6 weeks and the PCS was heparinized with 5ml of heparinized serum 100U/ml every 6 weeks. When accessing the PCS aseptic techniques were followed: local asepsis with betadine or chlorhexidine, sterile glows, disinfection of connections with chlorhexidine, transparent sterile dressing. The PCS dressing was inspected every day and if there were any changes the dressing was replaced. If there were no changes in the dressing it must have been changed every 7 days. When the needle was removed a sterile adhesive dressing was applied. Patients were educated to visually inspect the port every day and to contact the physician if there were any changes, pain or erythema.
- The current analysis primarily compares “early” versus “late” complications, and Table 3 performs subgroup comparisons only within the complication cases (N=25), which substantially limits statistical power. The borderline result for thrombosis (p = 0.051) should be interpreted cautiously and should not be overstated.
We thank the reviewer for the comment. The small number of patients included in the study and the small number of patients with complications is one of the limitations of the study and the results presented in this article indeed must be interpreted with caution. A paragraph with the limitations of this study was included in the Discussion section.
Even though this research provides a real-world picture of PCS-related complications in oncological patients it is important to acknowledge the limitations of this study. The retrospective nature of the study implies an increased risk of selection bias and lack of control over confounding factors such as insertion technique, level of experience of the personnel, or adherence to PCS care protocols. In addition this was a non-randomized study conducted in a single institution. The limited number of patients included in this study, especially patients with PCS-related complications, weakens the statistical power of the analysis and may not present a complete characterization of the general population evaluated in the study.
- The study period is 2024, whereas the ethics approval date is 2025, and the manuscript states that all patients provided informed consent. The authors should clarify whether this is a prospective follow-up cohort or a retrospective chart review. If retrospective, they should state whether consent was waived, obtained retrospectively, or otherwise addressed according to institutional requirements.
We thank the reviewer for the time devoted to evaluating our manuscript. We would like to clarify why the study period is reported as 2024, while the ethics approval was granted in 2025.
As The Oncology Institute Prof. Dr. Ion Chiricuță Cluj-Napoca is also a teaching hospital, all patients undergoing treatment and/or surveillance are required to sign a general informed consent form. This document collects various patient-related information and includes consent for participation in current and/or future research involving patients recorded in the Institute’s database. The Ethics Committee reviews these consent forms as part of its assessment of the study protocol.
- The manuscript contains multiple spelling and grammatical errors; comprehensive language editing is recommended.
We thank the reviewer for the suggestion. Spelling and grammar check was conducted.
Round 2
Reviewer 1 Report
Comments and Suggestions for Authors
Although authors partially responded and improved revised manuscript, several issues remain.
- Figure 2 needs statistical analysis.
- Relationships between cancer type and thrombosis or cancer type and infection may be interesting.
- There were a few finding.
- Discussion and conclusion are not sufficient.
Author Response
Letter to reviewer 1
Manuscript ID: cancers-4049073
Complications of port-a-cath systems: an institutional study on Romanian oncological patients
Adina NemeÈ™ 1,2, Sebastian-Alexandru Pocol 3, Tunde Banciu 2 and Diana Voskuil-GaloÈ™ 1,2,*
Reviewer 1 Comments and Suggestions for Authors – round 2
Although authors partially responded and improved revised manuscript, several issues remain.
- Figure 2 needs statistical analysis.
We thank the reviewer for the suggestion. Statistical analysis of the data depicted in Figure 2 and description was added to the figure and figure legend.
- Relationships between cancer type and thrombosis or cancer type and infection may be interesting.
We thank the reviewer for the suggestion. Information about cancer type and infection/thrombosis was included in the Results section.
90% of the patients (9 patients) that presented infection as a PCS-related complication were patients diagnosed with colorectal cancer and 1 patient was diagnosed with breast cancer. Among the patients that presented as a complication thrombosis, 50% (3 patients) were diagnosed with colorectal cancer and 50% (3 patients) with breast cancer.
- There were a few finding.
- Discussion and conclusion are not sufficient.
We thank the reviewer for the time assigned to re-evaluate our reviewed manuscript. We appreciate the suggestions to further extend the discussion section of the article. Please find additional notes regarding incidence of complications, right- versus left- subclavian vein implantation of PCS and catheter-related thrombosis and the correlations between cancer type and catheter related thrombosis.
In a study that included 1716 patients Li et al showed a PCS-related complication incidence of 18.5% which is similar to the incidence reported in our study [7].
Ignatov et al showed in a study on 550 patients that the highest incidence of complications was observed in patients with a totally implanted central venous access port placed in the left subclavian vein or with the catheter tip localized in the peripheral part of the superior vena cava [13]. In a study on 1848 patients catheter occlusion was statistically more frequent when the PCS was implanted in the left subclavian vein versus the right subclavian vein (3.5% vs 1%, p=0.001) [14].
In addition to the catheter-related factors and the thrombogenic predisposition associated with malignancy, patients with certain cancer types may have an increased risk of PCS-related thrombosis. One small study identified a correlation between catheter thrombosis and specific cancer locations, particularly colon, breast and kidney [26]. Another study identified breast cancer as a tumor site associated with an increased thrombotic risk and reported bladder cancer as having the strongest association with catheter-related thrombosis [27]. A study conducted on 600 patients suggested that gastric and pancreatic cancers were significantly associated with thrombosis occurrence [28]. Moreover, given the well-established hypercoagulable state associated with pancreatic cancer, one study reported an incidence of catheter-related thrombosis of nearly 15% in patients with pancreatic cancer [29]. In our study half of the patients that presented catheter thrombosis as a PCS-related complication were diagnosed with colorectal cancer and half with breast cancer. However, despite these well-supported correlations between cancer type and catheter thrombosis, the small number of patients with thrombosis in the current study does not entitle us to suggest a direct link between tumor location and subsequent complications. Nevertheless, large scale studies are required in order to further evaluate such clinical associations.
Literature data also indicate that PCS-related thrombosis is a common late complication, typically occurring after 30 days and often presenting with mild clinical signs such as ipsilateral limb or neck pain and swelling, while a proportion of cases remain asymptomatic [8,9]. The pathophysiological mechanisms of catheter-related thrombosis may explain its delayed occurrence, as prolonged catheter presence promotes vessel hyperplasia and thrombus stabilization. In addition, catheter colonization and infection could further potentiate thrombosis through fibrin sheath formation and persistent thrombin activity [9]. The incidence of PCS related thrombosis varies between 2.8% and 12.8% and it is lower than the incidence presented by PICC [26-28]. Our study noted a 24% incidence in catheter thrombosis. The variability in reported incidence rates may be attributed to differences in study design and endpoints, as well as heterogeneity in patient populations, cancer types, and additional risk factors across studies.
Reviewer 2 Report
Comments and Suggestions for Authors
The quality of manuscripts has improved significantly.
Author Response
Letter to reviewer 2
Manuscript ID: cancers-4049073
Complications of port-a-cath systems: an institutional study on Romanian oncological patients
Adina NemeÈ™ 1,2, Sebastian-Alexandru Pocol 3, Tunde Banciu 2 and Diana Voskuil-GaloÈ™ 1,2,*
Reviewer 2 Comments and Suggestions for Authors – round 2
The quality of manuscripts has improved significantly.
We want to thank the reviewer for the time allocated to re-evaluate our manuscript. We are pleased to know that the amends made in order to improve the content of this article were appreciated and we address our gratitude for reviewing our work.
Round 3
Reviewer 1 Report
Comments and Suggestions for Authors
Authors fully responded my comments. I have no further comment.